# Purine Metabolism and Pyrimidine Metabolism Alteration Is a Potential Mechanism of BDE-47-Induced Apoptosis in Marine Rotifer *Brachionus plicatilis*

**DOI:** 10.3390/ijms241612726

**Published:** 2023-08-12

**Authors:** Sai Cao, Jiayi Wang, Xinye You, Bin Zhou, You Wang, Zhongyuan Zhou

**Affiliations:** 1College of Marine Life Science, Ocean University of China, Qingdao 266003, China; caosai@stu.ouc.edu.cn (S.C.); wangjiayi@ouc.edu.cn (J.W.); xinyeyou@outlook.com (X.Y.); zhoubin@ouc.edu.cn (B.Z.); wangyou@ouc.edu.cn (Y.W.); 2Laboratory for Marine Ecology and Environmental Science, Laoshan Laboratory, Qingdao 266237, China

**Keywords:** purine metabolism, pyrimidine metabolism, reactive oxygen species, DNA damage, apoptosis, BDE-47, *Brachionus plicatilis*

## Abstract

This present study was conducted to provide evidence and an explanation for the apoptosis that occurs in the marine rotifer *Brachionus plicatilis* when facing 2,2′,4,4′-tetrabromodiphenyl ether (BDE-47) stress. Metabolomics analysis showed that aminoacyl-tRNA biosynthesis, valine, leucine and isoleucine biosynthesis, and arginine biosynthesis were the top three sensitive pathways to BDE-47 exposure, which resulted in the reduction in the amino acid pool level. Pyrimidine metabolism and purine metabolism pathways were also significantly influenced, and the purine and pyrimidine content were obviously reduced in the low (0.02 mg/L) and middle (0.1 mg/L) concentration groups while increased in the high (0.5 mg/L) concentration group, evidencing the disorder of nucleotide synthesis and decomposition in *B. plicatilis*. The biochemical detection of the key enzymes in purine metabolism and pyrimidine metabolism showed the downregulation of Glutamine Synthetase (GS) protein expression and the elevation of Xanthine Oxidase (XOD) activity, which suggested the impaired DNA repair and ROS overproduction. The content of DNA damage biomarker (8-OHdG) increased in treatment groups, and the p53 signaling pathway was found to be activated, as indicated by the elevation of the p53 protein expression and Bax/Bcl-2 ratio. The ROS scavenger (N-acetyl-L-cysteine, NAC) addition effectively alleviated not only ROS overproduction but also DNA damage as well as the activation of apoptosis. The combined results backed up the speculation that purine metabolism and pyrimidine metabolism alteration play a pivotal role in BDE-47-induced ROS overproduction and DNA damage, and the consequent activation of the p53 signaling pathway led to the observed apoptosis in *B. plicatilis.*

## 1. Introduction

Polybrominated Diphenyl Ethers (PBDEs) remain the most widely used brominated flame retardants in the world despite being recognized as persistent organic pollutants (POPs) and restricted by the Stockholm Convention for several years, and their impacts on ecosystems continue to be a major concern worldwide [1]. The oceans serve as the ultimate sink for global pollutants. PBDEs released from products are introduced into the ocean via atmospheric deposition and surface runoff or directly through electronic waste and plastic waste discarded into the ocean [1]. As a result of their lipophilic properties, PBDEs can readily accumulate in marine organisms upon entering the ocean and induce toxic effects [2].

Rotifers are crucial zooplankton in marine ecosystems, and the species *Brachionus plicatilis* is widely used as a model organism for toxicity assessment for marine ecosystems [3]. Our previous study has systematically assessed the toxic effects of 2,2′,4,4′-tetrabromodiphenyl ether (BDE-47), the most biotoxic congener of PBDEs on *B. plicatilis*, and preliminarily elucidated the mechanism of toxicity. The obtained results proved that BDE-47 exposure caused a decrease in the population density and reproductive capacity of *B. plicatilis* [4,5]. The toxic mechanism was discussed from the view of reactive oxygen species (ROS); BDE-47 induced ROS overproduction and inhibited the activity of the antioxidant system, which resulted in oxidative stress and mediated ovarian cell autophagy and apoptosis and eventually led to reproductive toxicity [6,7,8]. However, ROS was not an activator for the apoptosis pathway. It should act on specific targets such as inducing DNA oxidative damage, protein degeneration, and biofilm structure destruction, which, in turn, activate upstream signaling pathways like p53 and MAPK to mediate apoptosis [9]. Here brings the research interest of this present study: Did and how did ROS overproduction induce apoptosis in *B.plicatilis* exposed to BDE-47? Where did the excessive ROS in *B. plicatilis* come from?

Metabolomics can reflect life activity changes in a particular situation by detecting the end products of all biological pathways and metabolic processes, which provides new methods to reveal the toxic effects and mechanisms of environmental pollutants on marine organisms [10]. For example, the metabolomics results of Fu et al. [11] indicated that the inhibition of prostaglandin synthesis and the changes in the carnitine shuttle pathway were important reasons for the rapid accumulation of diclofenac in *Hyalella azteca*, which provided complementary insights to the toxicokinetic processes of diclofenac in *H. azteca*. Huang et al. [12] found that the metabolic disorder of tyrosine, phenylalanine, histidine, and beta-alanine was the potential mechanism of microplastics-induced oxidative stress and neurotoxicity in marine mussels *Mytilus coruscus* via metabolomics analysis combined with biochemical detection. Therefore, in order to use rotifers to evaluate the marine ecological risk of BDE-47 more effectively and clearly, the metabonomic analysis combined with biochemical detection was used in this research to elucidate the complete mechanism of BDE-47-induced cell apoptosis in *B. plicatilis* from the perspective of metabolic changes.

## 2. Results

### 2.1. Metabolomics Analysis of B. plicatilis with BDE-47 Stress

The volcano plots were depicted based on the number of identified differential metabolites of *B. plicatilis* (Figure 1A). The significantly different metabolites in low (0.02 mg/L) and medium (0.1 mg/L) concentration groups were obviously more than those in the high-concentration (0.5 mg/L) group, among which downregulated differential metabolites (FC < 0.5, *p* < 0.05) accounted for a large proportion in low- and medium-concentration groups, while the upregulated ones (FC > 2, *p* < 0.05) were generally found in high concentration groups. The results of the Principal Component Analysis (PCA) showed that the low- and medium-concentration groups were obviously separated from the control group, but the high-concentration group overlapped with it (Figure 1B), suggesting that the metabolome overall difference of *B. plicatilis* in low- and medium-concentration groups was greater than that in the high-concentration group. The significantly different metabolites in each treatment group were subjected to metabolic pathway enrichment analysis (Figure 1C). In the first five influenced pathways, aminoacyl-tRNA biosynthesis, valine, leucine, and isoleucine biosynthesis, arginine biosynthesis belonged to the category of amino acid metabolism, the pyrimidine metabolism and purine metabolism belonged to the category of nucleotide metabolism.

The fold change in metabolites in the above pathways compared with the control group was analyzed in the form of a heat map (Figure 2). Fourteen kinds of amino acids, including L-Histidine, L-Leucine, and L-Valine, etc., and three kinds of amino acid derivatives L-Asparagine, L-Glutamine, and ketoisocaproic acid showed an overall downregulated trend in the low and medium concentration groups. Moreover, L-Histidine was significantly downregulated (*p* < 0.05), L-Leucine and L-Valine were significantly upregulated (*p* < 0.05), and the remaining amino acids were not statistically different in the high-concentration group. The results suggested that low and medium concentrations of BDE-47 stress reduced the overall level of the amino acid pool in *B. plicatilis*. Changes in nucleotide metabolites in the pyrimidine metabolism, and purine metabolism pathways were presented and analyzed with clearer network diagrams in Section 2.2.

### 2.2. BDE-47 Impacted the Purine Metabolism and Pyrimidine Metabolism Pathways of B. plicatilis

Eligible differential metabolites (*p* < 0.05) in pyrimidine metabolism and purine metabolism pathways of *B. plicatilis* were screened and drawn into a network diagram (Figure 3). In pyrimidine metabolism, the pyrimidine nucleosides and Uracil downregulated significantly, and Thymine and Cytosine had no significant changes in low- and medium-concentration groups. Both pyrimidine nucleosides and bases upregulated significantly in high-concentration groups. In purine metabolism, Guanine, Hypoxanthine, Xanthine, and their corresponding nucleosides downregulated significantly, while Adenine, Adenosine, and Uric Acid had no difference in the low- and medium-concentration groups. Adenine and Adenosine upregulated significantly, Uric Acid downregulated significantly, and other metabolites had no significant changes in high-concentration groups. In short, the nucleotide metabolites in low- and medium-concentration groups showed a downward trend, while those in high-concentration groups showed an upward trend.

In addition, L-Glutamine, the raw material for nitrogen supply in the nucleotide de novo synthesis pathway, and its precursor L-Glutamate were significantly reduced in low and medium concentration groups. The protein expression of Glutamine Synthetase (GS) was also significantly reduced (Figure 4A). The trends of Glutamate, Glutamine, and GS were consistent with those of nucleotide metabolites. Therefore, it can be speculated that the decrease in raw material and synthetase is one of the reasons for the decrease in nucleotide metabolites in rotifers with low and medium concentrations of BDE-47 stress. However, the purine metabolite Uric Acid was not reduced. This was associated with changes in the activity of Xanthine Oxidase (XOD), of which its catalytic activity was found to be significantly elevated with low and medium concentrations of BDE-47 stress (Figure 4B). Thus, Uric Acid had no changes under general purine downregulation. Moreover, the GS protein expression decreased significantly in the high-concentration group (*p* < 0.05), and the XOD enzyme activity was higher than the control group but had no statistical difference (*p* > 0.05). It should be noted that the reasons for the increase in some nucleotide metabolites and the decrease in Uric Acid in high-concentration groups were unclear and required further studies. In addition, the XOD-catalyzed production of H_2_O_2_ and O_2_^-^ is an important component of ROS. Therefore, we conducted further research on ROS.

### 2.3. BDE-47-Induced ROS Overproduction Caused DNA Damage and p53 Signaling Pathway Activation in B. plicatilis

The fluorescence intensity of the DCFH-DA probe in each treatment group increased significantly (*p* < 0.05) (Figure 5A,B), demonstrating the increasing level of ROS in *B*. *plicatilis*. At the same time, the content of 8-hydroxy-2′-deoxyguanosine (8-OHdG) was also significantly increased (*p* < 0.05) and showed a clear BDE-47 dose–effect relationship (Figure 5C). The addition of ROS scavenger NAC eliminated the excessive ROS in each treatment group (Figure 5B), and the content of 8-OHdG also recovered to no significant difference (*p* > 0.05) with the control group (Figure 5C). This proved that the overproduction of ROS under BDE-47 stress was the main cause of oxidative DNA damage in *B*. *plicatilis*.

When cells undergo DNA damage, the p53 signaling pathway is activated to perform DNA repair or apoptosis. The protein expression of transcription factor p53 increased continuously under BDE-47 stress (Figure 6A), with medium- and high-concentration groups showing significantly higher expressions than the control group (*p* < 0.05). The expression of the pro-apoptotic gene Bax was upregulated, while the anti-apoptotic gene Bcl-2 was downregulated after p53 activation. The ratio of Bax to Bcl-2 expression can reflect the activation degree of the apoptotic pathway. In this study, Bax protein expression was significantly increased in high-concentration groups (Figure 6C), Bcl-2 protein expression was significantly decreased in medium- and high-concentration groups (Figure 6D), and Bax/Bcl-2 ratio was significantly increased in medium- and high-concentration groups (Figure 6B). The results indicated that medium and high concentrations of BDE-47 stress activated the transcription factor p53, which, in turn, activated the apoptotic pathway of *B*. *plicatilis*. After the addition of NAC, the protein expression of p53 in each treatment group was significantly decreased (*p* < 0.05), the Bax protein expression and Bax/Bcl-2 ratio were significantly decreased in high-concentration groups (*p* < 0.05), while the rest did not change significantly compared with those before NAC addition (*p* > 0.05). These changes demonstrated that excessive ROS mediated the p53 and apoptotic pathways activation in *B*. *plicatilis*.

## 3. Discussion

In this study, metabolomics analysis showed that BDE-47 mainly impacted the aminoacyl-tRNA biosynthesis, valine, leucine, and isoleucine biosynthesis, pyrimidine metabolism, purine metabolism, and arginine biosynthesis pathways of marine rotifers *B*. *plicatilis*. Among them, the impacts on aminoacyl-tRNA biosynthesis, valine, leucine, and isoleucine biosynthesis, and arginine biosynthesis pathways led to the reduction in the amino acid pool level in *B*. *plicatilis*, which may impede the protein synthesis of rotifers. Regarding the impacts on purine metabolism and pyrimidine metabolism pathways, specifically, the majority of metabolites were significantly downregulated in low- and medium-concentration groups and upregulated in the high-concentration group, along with significant changes in GS and XOD. Notably, a considerable number of toxicological studies in model organisms such as drosophila and mice have reported a sensitive response of purine metabolism and pyrimidine metabolism to BDE-47 stress. For example, Ji et al. [13] reported significant changes in Deoxyguanosine, Inosine, Uridine, 5-Methylcytidine, and other purine and pyrimidine metabolites in drosophila after BDE-47 stress, with a highly significant elevation of 5-Methylcytidine indicative of DNA methylation. Yang et al. [14] reported that BDE-47 caused a significant increase in purine metabolites such as ATP, IMP, Hypoxanthine, and Uric Acid in the blood and adipose tissue of mice on a high-fat diet, and oxidative stress was detected. Wei et al. [15] found that the metabolic pathways most affected by BDE-47 in human breast cancer cells were purine metabolism and pyrimidine metabolism, and the associated metabolites were significantly down-regulated. The results of this study were consistent with the above reports and demonstrated that the sensitive response mechanism of purine metabolism and pyrimidine metabolism to BDE-47 stress was also present in the marine rotifer.

Purine metabolism and pyrimidine metabolism are collectively known as nucleotide metabolism. Nucleotides are vital metabolites in organisms and participate in almost all biochemical processes in cells. The most important biological function of nucleotides is as the precursor for DNA and RNA biosynthesis [16]. Therefore, the changes in nucleotide metabolites are closely related to the synthesis and degradation of nucleic acids. In this study, the reduction in Glutamine, Glutamate, and GS in *B*. *plicatilis* caused by BDE-47 stress at low and medium concentrations was undoubtedly the direct cause of the reduction in nucleotide metabolites. Because all de novo nucleotide synthesis requires Glutamine for nitrogen supply, GS can catalyze the generation of Glutamine from Glutamate and ammonia [17]. In addition to the de novo synthesis pathway, organisms can also use free bases and nucleosides for salvage synthesis of the corresponding nucleotides [16]. When the de novo synthesis pathway was inhibited, rotifers consume free bases and nucleosides for salvage synthesis of nucleotides to meet the requirements of life activities [18]. This is most likely another reason for the reduction in purines, pyrimidines, and nucleosides in the low- and medium-concentration groups. However, a significant decrease in GS and a significant increase in nucleotide metabolites occurred in the high-concentration group. Fu et al. [17] found that GS expression was closely related to DNA repair ability, and GS knockout significantly inhibited the repair of DNA damage in cancer cells. Similarly, we found that high concentrations of BDE-47 caused a decrease in GS expression and severe DNA damage in *B*. *plicatilis*, indicating an impaired DNA repair capacity. A large amount of damaged DNA was then degraded to generate bases and nucleosides, resulting in an increase in nucleotide metabolites in the high-concentration group.

The changes in purine metabolism and pyrimidine metabolism may affect a series of life activities of *B*. *plicatilis*. Firstly, the impaired nucleotide synthesis pathway inevitably affected the available quantity of energy sources such as ATP and GTP. Our previous study on mussel hemocytes demonstrated this: the total adenosine pool (including ATP, ADP, and AMP) content of hemocytes was significantly reduced after BDE-47 stress [19]. Meanwhile, our latest study also proved that BDE-47 induced a decrease in the total adenosine content and energy reserve of *B*. *plicatilis* (unpublished data). The reduction in available energy may lead to the inhibition of rotifers’ growth and reproduction, which was also demonstrated in our previous study [5,6]. Secondly, the Xanthine Oxidase (XOD)-catalyzed production of H_2_O_2_ and O_2_^−^ has been shown to be the main source of excess ROS in cells under BDE-47 stress [20]. Similarly, the activation of XOD in purine metabolism promoted ROS overproduction of *B*. *plicatilis* in this study, while the collapse of the antioxidant system and the oxidative damage caused by BDE-47-induced excessive ROS production has been widely reported [6,21,22,23].

Among the effects of purine metabolism and pyrimidine metabolism changes, the most interesting to us was ROS overproduction. Our previous study demonstrated that ROS overproduction activated the ovary mitochondrial apoptotic pathways of *B*. *plicatilis* [7]. However, according to existing knowledge, ROS cannot directly activate the apoptotic pathway. Therefore, we further investigated the molecular mechanism of ROS-mediated rotifer apoptosis in this study. ROS is a recognized mediator of DNA damage [9,24]. The transcription factor p53 is activated after DNA damage. Under the regulation of p53, the cell cycle is arrested, and DNA repair is performed. When DNA damage is too severe to repair, p53 upregulates pro-apoptotic genes such as Bax to initiate apoptosis to prevent defective DNA replication [25]. In this study, the addition of the ROS scavenger NAC helped us to demonstrate that BDE-47-induced ROS overproduction mediated DNA damage and p53 activation in *B*. *plicatilis*. Unfortunately, the reduction in GS, Glutamine, and other nucleotide metabolites prevented DNA repair. Thus, the Bax/Bcl-2 ratio was continuously increased, and the apoptotic pathway was initiated. In summary, we suggested that the decrease in metabolites and key enzyme (GS, XOD) activities in purine metabolism and pyrimidine metabolism were crucial molecular mechanisms of BDE-47-induced apoptosis in *B*. *plicatilis.*

The adverse outcome pathway (AOP) is a conceptual framework applied to ecotoxicological study and risk assessment [26]. It describes adverse outcomes (AO) at the individual and population levels of organisms triggered by molecular initiation events (MIE) after chemical stress, which contain a series of key events (KE) at the cell and tissue levels [27]. Integrating our previous series of studies [5,6,7] and the results of this study, we completely described an AOP of BDE-47 on *B*. *plicatilis* (Figure 7). It took the change in key enzymes (GS, XOD) as MIE and DNA damage activated ovary cell apoptosis constituted a series of KE, which ultimately led to the AO of individual reproductive inhibition and population density reduction. This AOP provided important toxicological information for the marine ecological risk assessment of PBDEs.

## 4. Materials and Methods

### 4.1. Organism and Chemicals

The *B. plicatilis* have been maintained in our laboratory for several generations. They were cultured in beakers (2L) with sterilized seawater in an illuminating incubator (GHP-500E, Shanghai Sanfa, CHN). The culture system was maintained at 25 ± 1 °C and 60 μmol photon/(m^2^·s) with a 12h/12h light/darkness cycle. The seawater in each beaker was replaced once daily, and the *Chlorella* sp. (Chlorophyta) with 1.0 × 10^6^ cells/mL density was used as bait. Healthy amictic females were selected for subsequent experiments.

BDE-47 (GC-MS, >99.99% purity, white solid powder) was purchased from AccuStandard (New Haven, CT, USA) and dissolved in Dimethyl sulfoxide (DMSO, GC grade, >99.0%, liquid; Sigma-Aldrich, St. Louis, MI, USA) to prepare the stock solution of 2000 mg/L. The stock solution was diluted to working solution of required concentrations with sterilized seawater. According to our previous study [6], the concentration of the cosolvent DMSO used in this study (0.5%, *v*/*v*) was well below the no observed effect concentration (NOEC 13.5%, *v*/*v*) on *B. plicatilis*.

### 4.2. Experimental System

The *B. plicatilis* was exposed to BDE-47 in beakers with a density of 100 ind/mL. Each beaker contains 5000 individuals. The working concentrations of BDE-47 were set at 0.02 mg/L (low concentration group), 0.1 mg/L (medium concentration group), and 0.5 mg/L (high concentration group), respectively, according to our previous studies [6,7]. Group without BDE-47 addition was set as the blank control. Each treatment was conducted in independent triplicate. The rotifers were filtered via 400 mesh sieve silk and collected for subsequent analysis after 24h of exposure.

### 4.3. Metabolomics Analysis

Biomass at 0.2 g (about 10,000 individuals) of *B. plicatilis* was collected and added with 200μL ultrapure water and 800 μL methanol/acetonitrile (1:1, *v*/*v*) to constitute a sample. Each treatment contained 10 replicates samples for metabolomics analysis. The samples were disrupted with ultrasound at low temperatures and incubated at −20 °C for 1h to precipitate proteins. Then, the samples were centrifuged at 14,000 rpm and 4 °C for 15min. The supernatant was lyophilized and stored at −80 °C for metabolomics analysis. The samples were analyzed using ultrahigh performance liquid chromatography (UHPLC, 1290 Infinity LC, Agilent, Santa Clara, CA, USA) combined quadrupole-time of flight (Q-TOF) mass spectrometry (Triple TOF 5600+, AB SCIEX, Framingham, MA, USA) system. The metabolites were separated via hydrophilic interaction chromatography column (HILIC, 1.7 µm, 2.1 × 100 mm, Waters, Milford, MA, USA) and detected using electrospray ionization (ESI) in both positive and negative ionization modes. Detailed parameters and procedures of LC-MS were shown in Appendix A.

The raw data of metabolomics were converted into mzXML format using ProteoWizard. XCMS program was used to perform peak alignment, retention time correction, and peak area extraction, and the ion peaks with missing values >50% in the group were deleted. Metabolites were identified using accurate mass number (<25 ppm) and secondary spectrogram matching method and searched in the self-built database. SIMCA-P14.1 (Umetrics, Umea, Sweden) was used for pattern recognition. After preprocessing via Pareto-scaling, the data were statistically analyzed. Multidimensional statistical analysis included Principal Component Analysis (PCA). Unidimensional statistical analysis included Student’s *t*-test and Fold Change (FC) Analysis. Metabolites with Variable Importance for Projection (VIP) > 1 and *p* < 0.05 were considered significantly different. The volcano plot and heat map were made using R software. The metabolic pathways enrichment analysis of differential metabolites was performed in KEGG database (www.kegg.jp (accessed on 14 July 2022)) and MetaboAnalyst (www.metaboanalyst.ca (accessed on 14 July 2022)).

### 4.4. Determination of Key Enzyme in Purine Metabolism and Pyrimidine Metabolism

GS catalyzes glutamate and ammonia to synthesize Glutamine that is one of the important precursors of de novo nucleotide synthesis XOD is a key enzyme in purine metabolism, catalyzing the conversion of Hypoxanthine to Xanthine and the generation of Uric Acid, hydrogen peroxide, and superoxide anion. Biomass at 0.2 g (about 10,000 individuals) of *B. plicatilis* was collected into each sample and added with 2 mL PBS solution, and the mixture was completely disrupted by an ultrasonic cell disruption (JY92-IIX, Ningbo Scientz, Ningbo, China). The homogenate was centrifuged for 20 min (2500 rpm, 4 °C), and the supernatant was collected for further determination. GS protein expression and XOD activity were measured via Elisa kit (Shanghai Jining, Shanghai, China) and Colorimetric assay kit (Nanjing Jiancheng, Nanjing, China) following the manufacturer’s instructions, respectively. GS protein expression and XOD activity were expressed as per mg soluble protein (U/mg.pro.), and the total soluble protein content was measured via Bicinchoninic Acid protein assay kit (Shanghai Beyotime, Shanghai, China) to standardize GS protein expression and XOD activity. Each treatment was repeated in triplicates.

### 4.5. Determination of ROS, 8-OHdG Content, and p53 Signaling Pathway Related Proteins Expression with and without NAC Addition

Reactive oxygen species (ROS) level was measured using DCFH-DA probe (Nanjing Jiancheng, Nanjing, China) referring to the method described by Wang et al. [7]. In total, 30 individuals of *B. plicatilis* were randomly selected into a 96-well plate and observed and photographed under the fluorescence microscope (IX51, Olympus, Tokyo, Japan). The Image-Pro Plus 6.0 was used to determine the fluorescence intensities. 8-hydroxy-2′-deoxyguanosine (8-OhdG) is a pivotal biomarker for measuring DNA damage [28]. The content of 8-OhdG was measured via Elisa kit (Nanjing Jiancheng, Nanjing, China) following the manufacturer’s instruction. The key proteins expression of p53 signaling pathway, including p53, Bax, and Bcl-2, were measured via Elisa kit (Shanghai Jining, Shanghai, China) following the manufacturer’s instruction. Sample preparation and total protein concentration determination were identical to the description in Section 4.4. ROS scavenger NAC was added to pretreat *B. plicatilis* in a final concentration of 0.5 mM for 1 h according to our previous study [29]. Then, the rotifers were treated with BDE-47, and the changes of ROS, 8-OHdG, and p53 signaling pathway proteins were determined according to the above methods.

### 4.6. Statistical Analysis

The mean value and standard deviation (mean ± SD) were calculated with the different replicates of each treatment (n = 3). The figures were generated using Sigmaplot 12.5. The differences between treatments and control were analyzed via SPSS 16.0. The distribution normality and homogeneity of variance were analyzed using Kolmogorov–Smirnov test and Levene’s test, respectively. The significance was analyzed using one-way analysis of variance (ANOVA) followed by Waller–Duncan comparison. The data that did not conform to normal distribution and homogeneity of variance were analyzed using Kruskal–Wallis test. The significant difference was set to *p* < 0.05.

## 5. Conclusions

BDE-47 stress induced a disorder of purine metabolism and pyrimidine metabolism in *B. plicatilis*. In the above pathways, the excessive ROS produced by the increase in XOD activity caused DNA damage, and the reduction in Glutamine and GS inhibited DNA repair, which together exacerbated the DNA damage and activated the p53 and apoptosis. The above results proved that the alteration in purine metabolism and pyrimidine metabolism was the potential molecular mechanism of BDE-47-induced apoptosis in *B. plicatilis*, which completed the adverse outcome pathway (AOP) of reproductive inhibition and provided toxicological information for the marine ecological risk assessment of PBDEs.

## Figures and Tables

**Figure 1 ijms-24-12726-f001:**
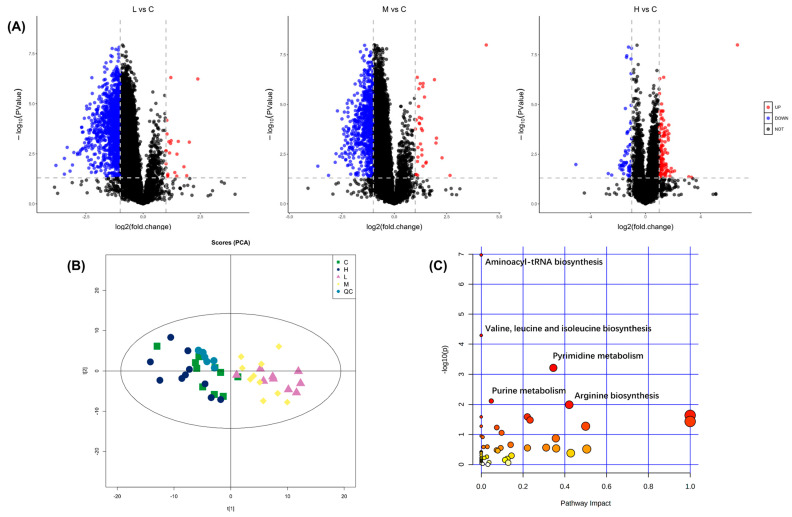
Metabolomics analysis of *B. plicatilis* exposed to BDE-47 for 24 h. (**A**) Volcano plots of different metabolites between low (0.02 mg/L, L), medium (0.1 mg/L, M), high (0.5 mg/L, H) concentration group, and control group (**C**). The blue dots represent significantly down-regulated metabolites (FC < 0.5, *p* < 0.05); the red dots represent significantly up-regulated metabolites (FC > 2, *p* < 0.05). (**B**) PCA score plots of each treatment group, control group, and quality control group. (**C**) Differential metabolic pathways enrichment analysis between each treatment group and control group. The metabolic pathways are displayed as distinctly colored circles depending on their enrichment analyses scores (vertical axis, shade of red) and topology (pathway impact, horizontal axis, circle diameter).

**Figure 2 ijms-24-12726-f002:**
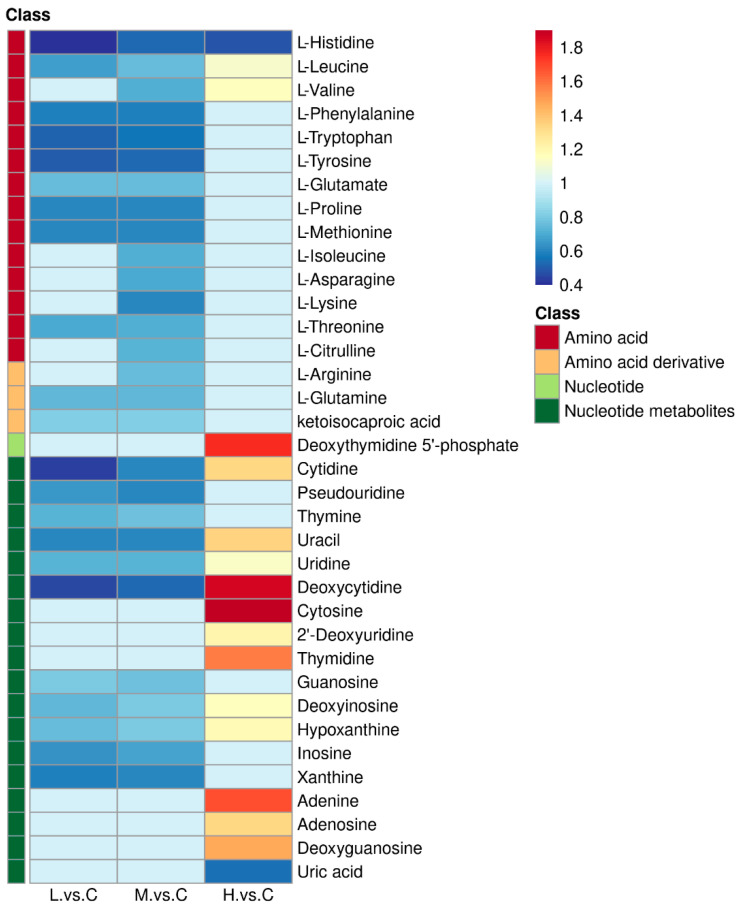
Heat map of differential metabolites fold change in aminoacyl-tRNA biosynthesis, valine, leucine, and isoleucine biosynthesis, pyrimidine metabolism, purine metabolism, and arginine biosynthesis pathways of *B. plicatilis* exposure to BDE-47. C: control group; L: low-concentration group (0.02 mg/L); M: medium-concentration group (0.1 mg/L); H: high-concentration group (0.5 mg/L).

**Figure 3 ijms-24-12726-f003:**
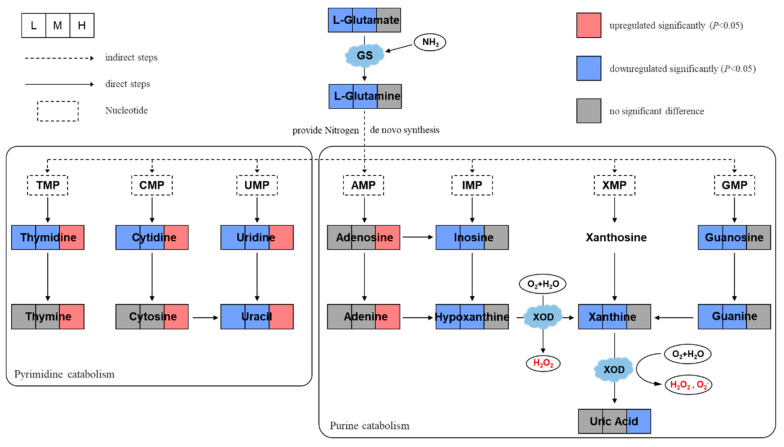
Network diagram of key metabolites in purine metabolism and pyrimidine metabolism pathways of *B. plicatilis* based on metabolomics analysis. The squares represent the changes of each metabolite in the low (0.02 mg/L, abbreviated as L), medium (0.1 mg/L, abbreviated as M), and high (0.5 mg/L, abbreviated as H) concentration group compared with the control group. GS: glutamine synthetase; XOD: xanthine oxidase.

**Figure 4 ijms-24-12726-f004:**
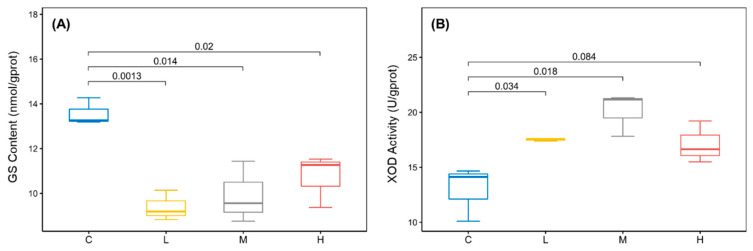
Changes in GS (**A**) and XOD (**B**) of *B*. *plicatilis* with BDE-47 stress. Different numbers represent *p*-values between groups. C: control group; L: low-concentration group (0.02 mg/L); M: medium-concentration group (0.1 mg/L); H: high-concentration group (0.5 mg/L).

**Figure 5 ijms-24-12726-f005:**
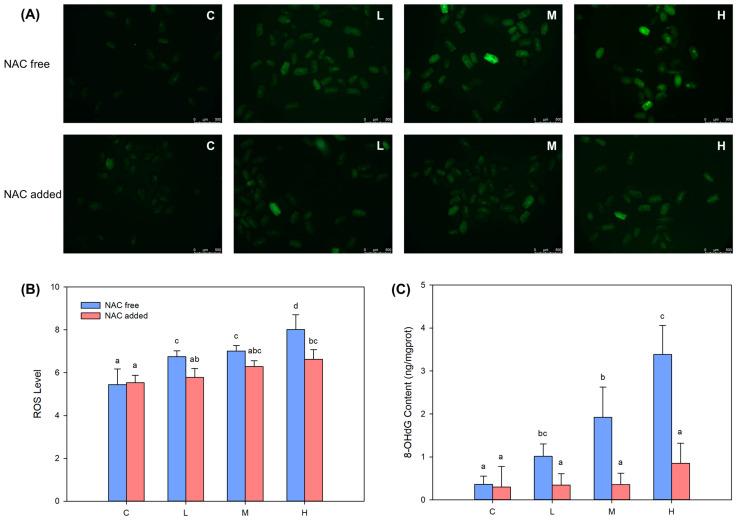
Changes of ROS level and 8-OHdG content in *B*. *plicatilis* with and without NAC addition with BDE-47 stress. (**A**) Fluorescence intensity photograph of ROS probe DCFH-DA. (**B**) Fluorescence intensity quantification of ROS probe DCFH-DA. (**C**) DNA damage biomarker 8-OHdG content. Different lowercase letters indicate significant differences (*p* < 0.05) between groups. C: control group; L: low-concentration group (0.02 mg/L); M: medium-concentration group (0.1 mg/L); H: high-concentration group (0.5 mg/L).

**Figure 6 ijms-24-12726-f006:**
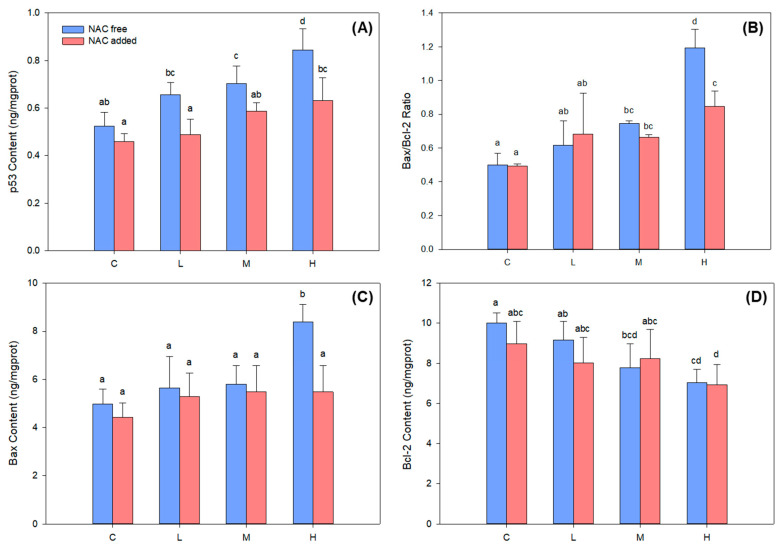
Changes of the key proteins in the p53 signaling pathway with and without NAC addition with BDE-47 stress in *B*. *plicatilis*. (**A**) Changes of p53 protein expression. (**B**) Changes in the ratio of Bax and Bcl-2 protein expression. (**C**) Changes of Bax protein expression. (**D**) Changes of Bcl-2 protein expression. Different lowercase letters indicate significant differences (*p* < 0.05) between groups. C: control group; L: low-concentration group (0.02 mg/L); M: medium-concentration group (0.1 mg/L); H: high-concentration group (0.5 mg/L).

**Figure 7 ijms-24-12726-f007:**
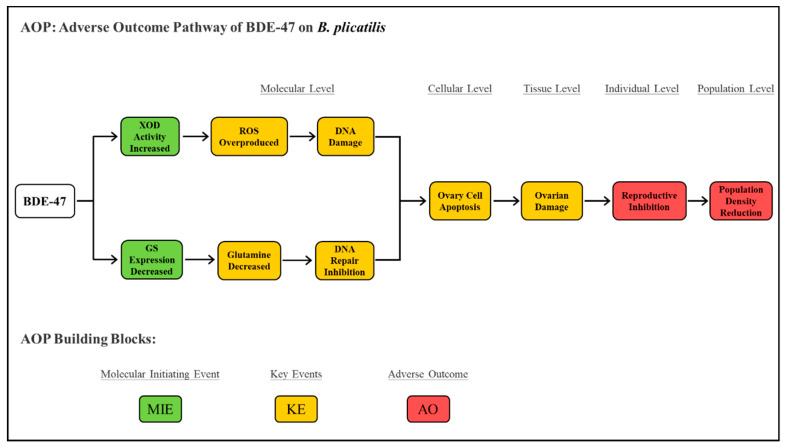
The adverse outcome pathway (AOP) of BDE-47 on *B. plicatilis.*

## Data Availability

Data are contained within this article.

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
