# Peer review of "Purine Metabolism and Pyrimidine Metabolism Alteration Is a Potential Mechanism of BDE-47-Induced Apoptosis in Marine Rotifer Brachionus plicatilis"

_ijms, 2023, doi:10.3390/ijms241612726_

Round 1

Reviewer 1 Report

MS ID: ijms-2509012

Title: Purine metabolism and pyrimidine metabolism alteration is a potential mechanism of BDE-47-induced apoptosis in marine rotifer Brachionus plicatilis

This manuscript contains research information on the apoptosis mechanism induced by BDE-47 in B. plicatilis. This study is valuable for better understanding the mechanism of BDE-47-induced apoptosis in marine rotifer. This research subject fits well with the scope of this journal. However, several points should be addressed and revised prior to final decision.     

 1. Section 2.2, Line 159: Authors suggested that GS content increased in the higher concentration than low and mid concentration. However, Figure 4 showed it was lower than that of control in all exposed concentration. Explain it.

 2. Section 4.2: Suggest the detail how many individuals of rotifer and replicates were used for exposure.

 3. Figure 7: In this study, authors investigated the apoptosis pattern using whole body of rotifer exposed to BDE-47. Thus, “ovary cell” apoptosis in cellular level of AOP seems to be overestimated, because any indexes related to reproductive inhibition in individual level did not observed in this study. 

Author Response

Thanks for your affirmation and suggestion, whih are valuable and helpful for improving our manuscript. All queries raised in the review comments are listed in the original order and answered one by one. The revised portion are marked in red in the manuscript. 

Query (1): Section 2.2, Line 159: Authors suggested that GS content increased in the higher concentration than low and mid concentration. However, Figure 4 showed it was lower than that of control in all exposed concentration. Explain it. 

Answer (1): Thanks for your suggestion. We have changed the clerical error "increased" to "decreased" and marked in red at line 159, which is consistent with the data in Figure 4.

Query (2): Section 4.2: Suggest the detail how many individuals of rotifer and replicates were used for exposure.

Answer (2):Thanks for your suggestion. We described the numbers of rotifer individual and the replicates for exposure at line 317 and 321 and marked in red. 

Query (3): Figure 7: In this study, authors investigated the apoptosis pattern using whole body of rotifer exposed to BDE-47. Thus, “ovary cell” apoptosis in cellular level of AOP seems to be overestimated, because any indexes related to reproductive inhibition in individual level did not observed in this study. 

Answer (3):Thanks for your suggestion. The AOP of BDE-47 on B. plicatilis in the manuscript was integrated based on the present study and our previous studies. The fluorescence staining results of our previous study showed that cells apoptosis and ROS overproduction of B. plicatilis mainly occurred in the ovary when exposured to BDE-47(Section 3.1 in A possible speculation on the involvement of ROS and lysosomes mediated mitochondrial pathway in apoptosis of rotifer Brachionus plicatilis with BDE-47 exposure. Doi:ARTN 147315 10.1016/j.scitotenv.2021.147315.). And the ovary damage led to the reproductive inhibition(Section 3.1 in The reproductive toxicity on the rotifer Brachionus plicatilis induced by BDE-47 and studies on the effective mechanism based on antioxidant defense system changes.  Doi:10.1016/j.chemosphere.2015.03.090.). We therefore linked ROS overproduction caused by disturbed purine and pyrimidine metabolism in rotifers to ovary apoptosis as a complete AOP.

Reviewer 2 Report

In my opinion, the peer-reviewed paper by Cao and co-authors (entitled: Purine metabolism and pyrimidine metabolism alteration is a potential mechanism of BDE-47-induced apoptosis in marine rotifer Brachionus plicatilis) fulfils the requirements for authors preempting publication in the International Journal of Molecular Science. I therefore recommend that the paper be accepted for publication in its present form. Congratulations to the authors.

The research submitted for review was conducted on a model of the sea rotifer (Brachionus plicatilis). The aim was to elucidate the mechanism of apoptosis induction under stress conditions induced by 2,2',4,4'-tetrabromodiphenyl ether (BDE-47) [BDE-47 marine ecological risk assessment].

The study used metabolomic analysis combined with biochemical detection and fluorescence visualisation.

Metabolomic analysis showed that aminoacyl-tRNA biosynthesis, valine, leucine and isoleucine biosynthesis and arginine biosynthesis were the three most sensitive pathways to BDE-47 exposure (reduced amino acid pool levels were observed). Purines and pyrimidines were markedly reduced in the low (0.02 mg / l) and medium (0.1 mg / l) concentration groups, while they increased in the high (0.5 mg / l) concentration group, indicating that nucleotide synthesis and degradation were impaired. There was a decrease in glutamine synthetase (GS) protein expression and an increase in xanthine oxidase (XOD) activity. Addition of NAC effectively reduced ROS overproduction, DNA damage and apoptosis. 

It was successfully confirmed that purine and pyrimidine metabolism plays a key role in BDE-47-induced ROS overproduction and DNA damage, and the resulting activation of the p53 signalling pathway led to the induction of apoptosis in B. plicatilis. In addition, an AOP was presented, providing important toxicological information for the ecological risk assessment of PBDEs in the marine environment.

Author Response

Thanks for your careful reading and affirmation. We hope everything goes well for you!

Reviewer 3 Report

The article makes a very high impression. The relevance of the work is due to the importance of environmental problems and pollution of world waters. In the article, the authors studied in detail and at a high scientific level the molecular mechanisms of metabolic changes under the influence of tetrabromodiphenyl ether on the marine rotifer. All methods are adequate and described in detail. I would like to note the high level of graphical presentation of the results, as well as an interesting and deep discussion. However, while reading, I noticed some minor design errors and I advise you to correct them.

11) In the Abstract, please, decipher NAC;

22)  In the keywords, give “reactive oxygen species” instead of ROS;

33) In the caption under Fig. 1 please decipher the meaning of C (...control group, C);

44) Some abbreviations are described twice or even more (GS — 151, 224, 353, and 397 lines, XOD — 157, 354 and 396 lines, NAC — 177 and 379 lines). I think it's redundant;

55)   Please check for typos (e.g., L-Valined), spaces, unnecessary capitalization.

Author Response

Thanks for your affirmation and suggestion, whih are valuable and helpful for improving our manuscript. All queries raised in the review comments are listed in the original order and answered one by one. The revised portion are marked in red in the manuscript. 

Suggestion (1): In the Abstract, please, decipher NAC .

Answer (1): Thanks for your suggestion. We have deciphered NAC in the abstract  and marked in red at line 33. 

Suggestion (2):In the keywords, give “reactive oxygen species” instead of ROS.

Answer (2): Thanks for your suggestion. We have replaced ROS with reactive oxygen species in the keywords and marked in red at line 39.

Suggestion (3): In the caption under Fig. 1 please decipher the meaning of C (...control group, C)

Answer (3): Thanks for your suggestion. We have added the meaning of C at line 106 and marked in red. 

Suggestion (4): Some abbreviations are described twice or even more (GS — 151, 224, 353, and 397 lines, XOD — 157, 354 and 396 lines, NAC — 177 and 379 lines). I think it's redundant.

Answer (4): Thanks for your suggestion. We have deleted the redundant description of abbreviations at line 151, 224, 353, 397, 157, 354, 396, 177 and 379.   

Suggestion (5): Please check for typos (e.g., L-Valined), spaces, unnecessary capitalization.

Answer (5): Thanks for your redress. We have corrected the typos at line 115 and 118.